# Genome-Wide Identification and Analysis of bHLH Transcription Factors Related to Anthocyanin Biosynthesis in *Cymbidium ensifolium*

**DOI:** 10.3390/ijms24043825

**Published:** 2023-02-14

**Authors:** Meng-Jie Wang, Yue Ou, Zuo Li, Qing-Dong Zheng, Yu-Jie Ke, Hui-Ping Lai, Si-Ren Lan, Dong-Hui Peng, Zhong-Jian Liu, Ye Ai

**Affiliations:** 1Key Laboratory of National Forestry and Grassland Administration for Orchid Conservation and Utilization, College of Landscape Architecture and Art, Fujian Agriculture and Forestry University, Fuzhou 350002, China; 2Guangdong Key Laboratory of Ornamental Plant Germplasm Innovation and Utilization, Environmental Horticulture Research Institute, Guangdong Academy of Agricultural Sciences, Guangzhou 510640, China

**Keywords:** *Cymbidium ensifolium*, bHLH transcription factor, anthocyanin biosynthesis, expression analysis

## Abstract

The basic helix-loop-helix (bHLH) transcription factors are widely distributed across eukaryotic kingdoms and participate in various physiological processes. To date, the bHLH family has been identified and functionally analyzed in many plants. However, systematic identification of bHLH transcription factors has yet to be reported in orchids. Here, 94 bHLH transcription factors were identified from the *Cymbidium ensifolium* genome and divided into 18 subfamilies. Most *CebHLHs* contain numerous *cis-acting* elements associated with abiotic stress responses and phytohormone responses. A total of 19 pairs of duplicated genes were found in the *CebHLHs*, of which 13 pairs were segmentally duplicated genes and six pairs were tandemly duplicated genes. Expression pattern analysis based on transcriptome data revealed that 84 *CebHLHs* were differentially expressed in four different color sepals, especially *CebHLH13* and *CebHLH75* of the S7 subfamily. The expression profiles of *CebHLH13* and *CebHLH75* in sepals, which are considered potential genes regulating anthocyanin biosynthesis, were confirmed through the qRT-PCR technique. Furthermore, subcellular localization results showed that CebHLH13 and CebHLH75 were located in the nucleus. This research lays a foundation for further exploration of the mechanism of *CebHLHs* in flower color formation.

## 1. Introduction

Transcription factors are a type of regulatory proteins that are mainly combined with the measuring component on gene promoters to activate or repress the expression of downstream target genes [1,2]. bHLH transcription factors are widely distributed in eukaryotes and are named for their highly conserved basic helix-loop-helix (bHLH) domain [3]. The bHLH domain consists of ≈60 amino acids with two functionally distinct regions, a basic region and a helix-loop-helix (HLH) region [4,5]. The basic region is predicted to recognize and bind DNA located at the N-terminus and contains 13–17 amino acids [2,6], typically with a highly conserved His5-Glu9-Arg13 motif [2,7,8]. The HLH region comprises two α-helices linked by a loop of variable length, located at the C-terminus, and consists of ≈40 amino acids. The bHLH proteins containing the HLH region are generally able to form homodimers or heterodimers with other proteins [9,10], which is a prerequisite for DNA recognition and DNA-binding specificity and is also an important factor affecting the expression pattern of bHLH transcription factors [2,7].

Numerous studies have shown that bHLH transcription factors play essential roles in the physiological process of plants, such as plant growth and development [11,12,13,14], abiotic stress response [15,16,17,18], and anthocyanin biosynthesis [19,20,21]. Lc was the first bHLH protein identified in plants, and it was shown to regulate anthocyanin biosynthesis in maize [22]. In orchids, the role of bHLH proteins in anthocyanin biosynthesis has been continuously explored. *DhbHLH1* interacts with *DhMYB2* to regulate the synthesis of anthocyanins in the petals of *Dendrobium* hybrids, while *DhbHLH1* can also independently regulate the synthesis of anthocyanins in the lips [23]. In *Paphiopedilum hirsutissimum*, *bHLH14* and *bHLH106* were found to be involved in the formation of flower color [24]. *DcTT8* regulates anthocyanin biosynthesis in the stems of *D. candidum*, which causes them to exhibit a red phenotype [25]. Three anthocyanin biosynthesis-regulating genes, *CybHLH1*, *CybHLH2*, and *CyMYB1*, were identified in *Cymbidium* ‘Mystique’ [26]. In addition, bHLH transcription factors can also interact with MYB and WD40 transcription factors to form an MBW complex to regulate the biosynthesis of anthocyanins. For example, simultaneous overexpression of *Lc* (a bHLH transcription factor) and *C1* (a MYB transcription factor) resulted in scarlet spots on the white petals of *C. hybrid* ‘Jung Frau dos Pueblos’ [27]. It was also found that *PlbHLH20*, *PlbHLH26*, *PlMYB10*, *PlWD40-1,* and their MBW protein complex in *Pleione limprichtii* may be involved in anthocyanin biosynthesis [28].

*Cymbidium ensifolium* is a traditional Chinese orchid with a long history of cultivation and high ornamental value [29]. Flower color is one of the main ornamental characteristics of *C. ensifolium*, which contains purple, dark red, yellow, green, white, and so on. However, the molecular mechanism of its formation remains unclear. Here, a total of 94 CebHLH transcription factors were identified, distributed in 18 subfamilies. They were named according to their location on the chromosomes, and a series of analysis of their characteristics, phylogenetic relationships, gene structure, conserved motifs, *cis-acting* elements, and collinearity analysis were conducted. In addition, the expression pattern of *CebHLHs* in four different color sepals were analyzed with transcriptome sequencing and qRT-PCR, screening out the potential genes *CebHLH13* and *CebHLH75* potentially involved in anthocyanin biosynthesis in *C. ensifolium*. This study lays a theoretical foundation for further exploring the role of *CebHLHs* in flower color formation.

## 2. Results

### 2.1. Identification of CebHLH Transcription Factors and Analysis of Their Physicochemical Properties

A total of 94 CebHLHs were identified from the *C. ensifolium* genome and named CebHLH1–CebHLH94 on the basis of their chromosomal locations. To further analyze the features of the 94 CebHLHs, we predicted their physicochemical properties. As shown in Appendix A, the amino acid lengths of CebHLHs ranged from 79 aa (CebHLH16) to 764 aa (CebHLH12), with an average length of 331 aa. The molecular weights (MW) of CebHLHs varied from 9.10 kDa (CebHLH16) to 83.82 kDa (CebHLH12), with an average MW of 36.55 kDa. The isoelectric point (pI) of these proteins ranged from 4.79 (CebHLH90) to 10.72 (CebHLH21). Furthermore, subcellular localization prediction revealed that most CebHLHs were located in the nucleus.

### 2.2. Phylogenetic Analysis of CebHLHs

To explore the function and evolutionary relationship of CebHLHs, a phylogenetic tree was constructed on the basis of the 94 CebHLH proteins and 152 AtbHLH proteins using the neighbor-joining (NJ) method (Figure 1). The results showed that all members were divided into 21 subfamilies, but CebHLH family members were distributed in 18 subfamilies (Figure 2A), which were absent in the S1, S11, and S14 subfamilies. Moreover, the S18 subfamily had the largest number of CebHLHs (13 members), followed by the S10 (10 members) and S19 subfamily (10 members), and the S12 and S21 subfamilies had the least number (with only one member). Notably, AtbHLH2, AtbHLH42, and AtbHLH12 in the S7 subfamily were previously found to be involved in the regulation of anthocyanin biosynthesis [30], and thus we speculated that CebHLH13 and CebHLH75 in the S7 subfamily may have similar functions and should be analyzed in the subsequent analysis.

### 2.3. Gene Structure and Motif Analysis of CebHLHs

Ten conserved motifs of CebHLHs were identified by the MEME program [31] (Figure 2B). Most CebHLHs contained three motifs, and members of the S7 and S8 subfamilies contain the highest number of motifs (8–9), whereas CebHLH16 contains only one. In addition, CebHLHs members in the same subfamily usually contain similar motifs, and some motifs only exist in the individual subfamily, which may be related to the evolution of gene functional diversity.

To understand the characteristics of the conserved domains of CebHLHs, the multiple sequence alignment results of 94 CebHLHs were uploaded to Weblogo [32]. As shown in Figure 3, the conserved bHLH domains constituted by motif 1 and motif 2 were highly conserved in the sequences of CebHLHs and contained 56 amino acids. The consensus of amino acid residues at 22 positions was higher than 50%, and the consensus of Arg-12, Leu-23, Pro-28, and Leu-39 was higher than 90%.

In order to further analyze the gene structure of *CebHLHs*, an intron–exon structure map of 94 *CebHLHs* was obtained (Figure 2C). There were significant differences in the number of introns or exons among *CebHLHs*. Among them, *CebHLH74* had the largest number of introns, with a total of nine introns. A total of 73.4% of CebHLHs contained 1–4 introns, while *CebHLH2*, *CebHLH3*, *CebHLH15*, *CebHLH20*, *CebHLH21*, *CebHLH31*, *CebHLH34*, *CebHLH36*, *CebHLH37*, *CebHLH45*, *CebHLH46*, *CebHLH47*, *CebHLH67*, *CebHLH78,* and *CebHLH93* did not contain introns, accounting for 16% of the total.

### 2.4. Promoter Analysis of CebHLHs

The promoter regions of *CebHLHs* contained a large number of *cis-acting* elements (Figure 4A), including light-responsive elements (GT1-motif), MeJA-responsive elements (TGACG motif and CGTCA motif), auxin regulatory elements (TGA-element), and abscisic acid regulatory elements (ABRE), among which light-responsive elements had the largest number (365), followed by MeJA response elements (310). All *cis-acting* elements were classified into three major categories (Figure 4B), namely, plant growth and development, abiotic stress responses, and phytohormone responses. Among these three categories, the abiotic stress response category had the largest number, followed by the phytohormone regulation category, while the number related to plant growth and development was relatively small. Notably, there were *cis-acting* elements (MBSI) in *CebHLH48* and *CebHLH54* that can bind to MYB and regulate flavonoid biosynthesis, suggesting that these two genes may be involved in flavonoid biosynthesis. The results of promoter analysis suggested that there were differences in the transcriptional regulation of *CebHLHs*, which may be related to their functional diversity.

### 2.5. Chromosomal Localization and Collinearity Analysis of CebHLHs

On the basis of the annotation information of the *C. ensifolium* genome, 94 *CebHLHs* were unevenly distributed on 19 chromosomes. Chromosome 3 contained the largest number of *CebHLHs* (12), followed by chromosomes 1 and 4 (9 each), while chromosome 20 had only one.

Gene duplication events are crucial evolutionary processes leading to gene structural and functional divergence [33]. A total of 19 pairs of duplicated genes were found in the CebHLH family, of which 6 pairs were tandemly duplicated genes (Figure 5), and 13 pairs were segmentally duplicated genes (Figure 6). Among the segmentally duplicated genes, *CebHLH25* and *CebHLH26* are located on chromosome 4, and *CebHLH45* and *CebHLH46* are located on chromosome 7. The largest number of tandemly duplicated genes was found on chromosomes 3 and 7, with three pairs, and only one was found on chromosomes 1, 6, 10, 13, and 20. Among the tandemly duplicated genes, *CebHLH20* and *CebHLH21*, *CebHLH4* and *CebHLH5*, *CebHLH50* and *CebHLH51*, and *CebHLH92* and *CebHLH93* share similar conserved motifs and gene structures. 

The nonsynonymous/synonymous mutation (*Ka/Ks*) ratio is essential for exploring genomic evolution [34], which can show purifying selection (*Ka/Ks* < 1), neutral mutation (*Ka/Ks* = 1), and positive selection (*Ka/Ks* > 1). As shown in Appendix A, the *Ka/Ks* ratios of 16 pairs of genes were between 0.11 and 0.54, indicating that these genes underwent strong purifying selection during evolution [35,36]. Notably, the *Ka/Ks* ratios of *CebHLH20* and *CebHLH21*, and *CebHLH4* and *CebHLH5* were greater than 1, suggesting that these two pairs of genes underwent positive selection [35,36].

### 2.6. Expression Pattern of CebHLHs in Four Different Color Sepals

To understand the expression patterns of *CebHLHs* and screen out potential genes that may be related to anthocyanin biosynthesis, we performed transcriptome sequencing using the sepals of four different colors of *C. ensifolium* (Appendix A). As shown in Figure 7, 82 genes had different expression levels in sepals of different colors and were divided into seven groups (A–G). The five genes in group A were lowly expressed in purple-red sepals and highly expressed in other sepals. Eight genes in group B were highly expressed in red sepals, among which *CebHLH75* was also expressed in purple-red sepals, but not expressed in yellow-green and white sepals, indicating that *CebHLH75* may regulate the biosynthesis of anthocyanins. Most genes in groups C and D were highly expressed in white sepals, and 14 genes in group E were mainly expressed in yellow-green sepals. In group F, five genes were highly expressed in purple-red and red sepals, and lower in yellow-green and white sepals. All genes in group G were highly expressed in purple-red sepals and lower in the other sepals. Notably, *CebHLH75* in group B and *CebHLH13* in group G, which belonged to the S7 subfamily, were highly expressed in red and purple-red sepals. Therefore, we speculated that *CebHLH75* and *CebHLH13* play a positive role in anthocyanin biosynthesis.

### 2.7. qRT-PCR Analysis of CebHLHs

qRT-PCR was used to further verify the expression patterns of *CebHLH13* and *CebHLH75* in four colored sepals (Figure 8). The results showed that the expression of *CebHLH13* was highest in purple-red sepals. *CebHLH75* had a high expression in purple-red and red sepals and was lowly expressed in yellow-green and white sepals. The expression levels of *CebHLH13* and *CebHLH75* in the four colored sepals obtained by qRT-PCR were consistent with the transcriptome data, supporting the accuracy of the transcriptome sequencing results.

### 2.8. Subcellular Localization of CebHLH13 and CebHLH75

To explore the subcellular localization of CebHLH13 and CebHLH75 proteins, two recombinant vectors (*35S: CebHLH13-GFP* and *35S: CebHLH75-GFP*) and the control vector (*35S: GFP*) were separately introduced into *Nicotiana benthamiana* leaves. The subcellular localization of these two proteins was observed after 48 h. As shown in Figure 9, the GFP signals of *35S: CebHLH13-GFP* and *35S: CebHLH75-GFP* were significantly detected in the nucleus, demonstrating that CebHLH13 and CebHLH75 were localized in the nucleus.

## 3. Discussion

The bHLH transcription factors are one of the largest transcription factor families in plants and play important roles in various physiological processes. However, the systematic identification of the bHLH transcription factor family in orchids has yet to be reported. Therefore, it is of great theoretical and practical significance to systematically classify and functionally study bHLH transcription factors of *C. ensifolium*. In this study, we identified 94 CebHLHs from the genome of *C. ensifolium*. The number was the same as *Vitis vinifera* (94 members) [37], less than *Arabidopsis thaliana* (162 members) [38], *Daucus carota* (146 members) [8], *Triticum aestivum* (225 members) [39], and *Solanum tuberosum* (124 members) [40], and more than *Paeonia suffruticosa* (84 members) [41] and *Citrus sinensis* (56 members) [42]. Differences in the number of bHLH family members are normal due to gene duplication, deletion, and functional diversification [43].

A total of 94 *C. ensifolium* bHLH proteins and 152 *A. thaliana* bHLH proteins were combined to construct a phylogenetic tree. According to the previous classification of *A. thaliana* [38], all members were divided into 21 subfamilies, of which the S1, S11, and S14 subfamilies did not contain CebHLHs, indicating that CebHLHs in these three subfamilies may have been differentiated or lost in the long-term evolutionary process. Proteins with similar functions tend to cluster in the same subfamily. In previous studies, AtbHLH2, AtbHLH42, and AtbHLH12 of the S7 subfamily were found to promote anthocyanin biosynthesis [30]. Therefore, it is speculated that CebHLH13 and CebHLH75 in the same subfamily also have the same function.

Previous studies have found that the bHLH domain consists of ≈60 amino acids, and its basic region has at least five highly conserved amino acids and a His5-Glu9-Arg13 conservative structure [2,6,7,8]; our study of CebHLHs supports this notion. The bHLH transcription factors often exert their biological functions by forming homologous or heterodimers [9,10], and Leu23 residues in the HLH region are crucial for dimer formation [8,30,44]. Sequence analysis found that the conservation of Leu23 of CebHLHs is as high as 99%, indicating that almost all CebHLHs can form homologous or heterodimers, which is of great significance for the biosynthesis of anthocyanins. In addition, gene structure and motif analysis found that the members of the same subfamily have similar gene structures. The specific conserved motifs in some subfamilies also support the reliability of the phylogenetic tree. Except for 15 *CebHLHs* that do not contain introns, the number of introns in most *CebHLHs* ranges from 1 to 9. The intron positions and numbers of *CebHLHs* are polymorphic, which may be the evolution of functional diversity during long-term evolution. Different *cis-acting* elements were observed in the promoter regions of *CebHLHs*, and the data showed that most of them were associated with abiotic stress responses, indicating that *CebHLHs* may be regulated by various factors and play an important role in the transcriptional regulation of abiotic stress responses.

Gene duplication events play a central role in the evolutionary process and are important for the generation of new gene members [33]. Our study revealed the gene duplication pattern of the CebHLH family and determined its relative location on the chromosomes. There were 19 pairs of repetitive genes in 94 *CebHLHs*, of which 13 pairs were segmentally duplicated genes, and 6 pairs were tandemly duplicated genes. In addition, 84.2% of the gene pairs had *Ka/Ks* ratios less than 1, suggesting that most *CebHLHs* underwent strong purifying selection during evolution [35,36]. However, we found two pairs of genes (*CebHLH20* and *CebHLH21*, *CebHLH4* and *CebHLH5*) with *Ka/Ks* ratios greater than 1, suggesting that these two pairs of genes underwent positive selection [35,36], which is of great significance to the study of species evolution.

Expression pattern analysis revealed that *CebHLH13* was highly expressed in the purple-red sepals but very low in the red sepals, which may be related to the type and content of anthocyanins it regulates. *CebHLH75* was significantly highly expressed in purple-red and red sepals, and lower in yellow-green and white sepals. Their expression levels were consistent with phenotype observation. Therefore, we speculated that *CebHLH13* and *CebHLH75* could promote anthocyanin biosynthesis in *C. ensifolium*. Previous research has shown that bHLH transcription factors play critical roles in anthocyanin biosynthesis. In this study, potential *CebHLHs* related to anthocyanin biosynthesis in *C. ensifolium* were screened out, laying a foundation for further exploring the mechanism of *CebHLHs* in flower color formation and also providing valuable information for orchid flower color breeding.

## 4. Materials and Methods

### 4.1. Plant Materials

*C. ensifolium* cultivars with purple-red sepals, red sepals, yellow-green sepals, and white sepals were collected from the Orchid Germplasm Resource Nursery of Fujian Agriculture and Forestry University, Fuzhou, Fujian Province, China. All materials were immediately frozen in liquid nitrogen and stored at −80 °C for later analysis. For each sample, three replicates were obtained from different plants.

### 4.2. Identification and Sequence Analysis of CebHLHs

The genome sequence and annotation information for *C. ensifolium* was downloaded from the National Genome Data Center (NGDC) (https://ngdc.cncb.ac.cn/, accessed on 25 March 2022). The HMM (Hidden Markov Model) profile of the bHLH domain (PF00010) was obtained from the Pfam database (http://pfam.xfam.org/search, accessed on 25 March 2022) for protein screening by TBtools software [45] (E-value ≤ 10^−4^). All protein sequences were further confirmed using SMART (http://smart.emblheidelberg.de/, accessed on 25 March 2022), and proteins without the bHLH domain were deleted. The physicochemical properties of the protein amino acid sequences of CebHLHs were predicted by ExPASy (http://www.expasy.org/tools/, accessed on 25 March 2022) [46], and subcellular localization prediction was performed on WoLF PSORT (https://wolfpsort.hgc.jp/, accessed on 25 March 2022) [47]. In addition, the multiple sequence alignment results of 94 CebHLH proteins were submitted to WebLogo (https://weblogo.berkeley.edu/logo.cgi, accessed on 25 March 2022) to observe their conserved domain characteristics [32].

### 4.3. Phylogenetic Analysis of CebHLHs

Multisequence alignment of 94 CebHLH proteins and 152 AtbHLH proteins was performed using MEGA11 and Jalview software [48,49], and the results were uploaded to MEGA11 to construct an unrooted neighbor-joining phylogenetic tree with 1000 bootstraps. Finally, the phylogenetic tree was imported into iTOL (https://itol.embl.de/itol.cgi, accessed on 25 March 2022) for modification [50].

### 4.4. Gene Structure and Motif Analysis of CebHLHs

The intron–exon structure of *CebHLHs* was identified by the Visualize Gene Structure program of TBtools [45]. Conserved motifs of CebHLHs were analyzed by MEME (https://meme-suite.org/meme/, accessed on 25 March 2022) [31]. The maximum number of motifs in the parameter settings was 10, while the other parameters were default. The results were then imported into TBtools for visualization [45].

### 4.5. Promoter Analysis of CebHLHs

The 2000 bp regions upstream of the start codon were extracted and uploaded to TBtools to identify the putative *cis-acting* elements in the promoter region [45]. Data processing was performed with Excel, and then the results were visualized with TBtools [45].

### 4.6. Chromosomal Localization and Collinearity Analysis of CebHLHs

After obtaining information on the location of the bHLH family on 19 chromosomes from the genome annotation information of *C. ensifolium*, the chromosomal location map was obtained with TBtools [45]. The collinear relationship between the chromosome pairs was drawn and visualized by the One Step MCScanx and Advance Circos program of TBtools [45]. Moreover, the *Ka/Ks* ratios were calculated by TBtools [45].

### 4.7. Expression Pattern of CebHLHs

Total RNA from four different color sepals was extracted using the OMEGA kit (Norcross, Georgia, USA). RNA-seq and library construction were performed by the Novogene Bioinformatics Co., Ltd. (Beijing, China) on an Illumina HiSeq 2500 platform. Expression levels of CebHLHs were represented by fragments per kilobase of exon model per million mapped reads (FPKM) values. The FPKM values of CebHLHs were imported into TBtools to generate a heat map [45].

### 4.8. qRT-PCR Analysis of CebHLHs

The PrimerScript^®^ RT Reagent Kit with gDNA Eraser (TaKaRa, Dalian, China) was used to reverse transcribe the RNA into cDNA. qRT-PCR was performed with three biological replicates and three technical replicates using Taq Pro Universal SYBR qPCR Master Mix (TaKaRa, Dalian, China) on an Applied Biosystems 7500 Real-Time System (Applied Biosystems, Foster City, CA, USA). All primers used for qRT-PCR are listed in Appendix A. *GAPDH* was used as an internal reference for the data. Relative expression was calculated using the 2^–∆∆CT^ method.

### 4.9. Subcellular Localization of CebHLH13 and CebHLH75

The full-length ORFs of *CebHLH13* and *CebHLH75* without the termination codon were inserted into the pCAMBIA1302 vector with *NcoI* and *SpeI* restriction sites to create the *35S: CebHLH13-GFP* and *35S: CebHLH75-GFP* fusion construct. In addition, the recombinant plasmid and control *35S: GFP* plasmid were transferred into *Agrobacterium* strain GV3101. The *Agrobacterium* containing the target plasmid was resuspended and transiently infected into *Nicotiana benthamiana* leaves. The subcellular localization of CebHLH13 and CebHLH75 were observed by LSM710 confocal laser microscopy (CarlZeiss, Jena, Germany) after 48 h. All primers used in this study are listed in Appendix A.

## 5. Conclusions

In this study, 94 CebHLHs were identified in the genome of *C. ensifolium* and subjected to classification, phylogenetic construction, gene structure analysis, conserved motif characterization, chromosomal localization, and expression pattern analysis. The expression patterns of these genes were specific in four colored sepals of *C. ensifolium*. Two potential genes, *CebHLH13* and *CebHLH75*, which may be related to anthocyanin biosynthesis, were screened. The research provided useful information for the functional analysis of bHLH transcription factors, as well as for flower color improvement and molecular breeding in orchids.

## Figures and Tables

**Figure 1 ijms-24-03825-f001:**
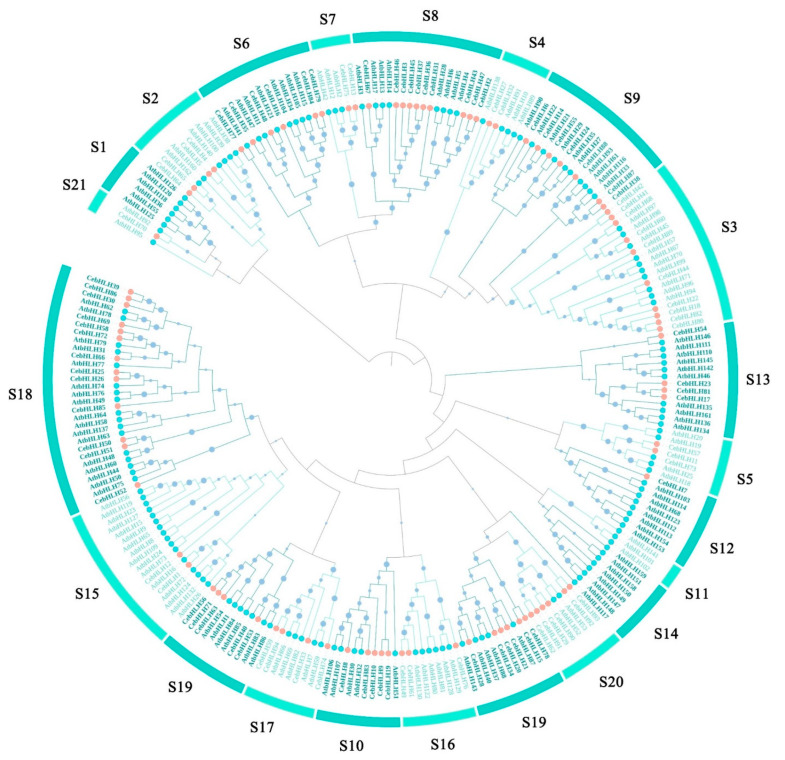
Phylogenetic tree of bHLH proteins based on 94 CebHLH proteins and 152 AtbHLH proteins. Orange circles represent CebHLHs, blue circles represent AtbHLHs. S1–S21 represent 21 subfamilies.

**Figure 2 ijms-24-03825-f002:**
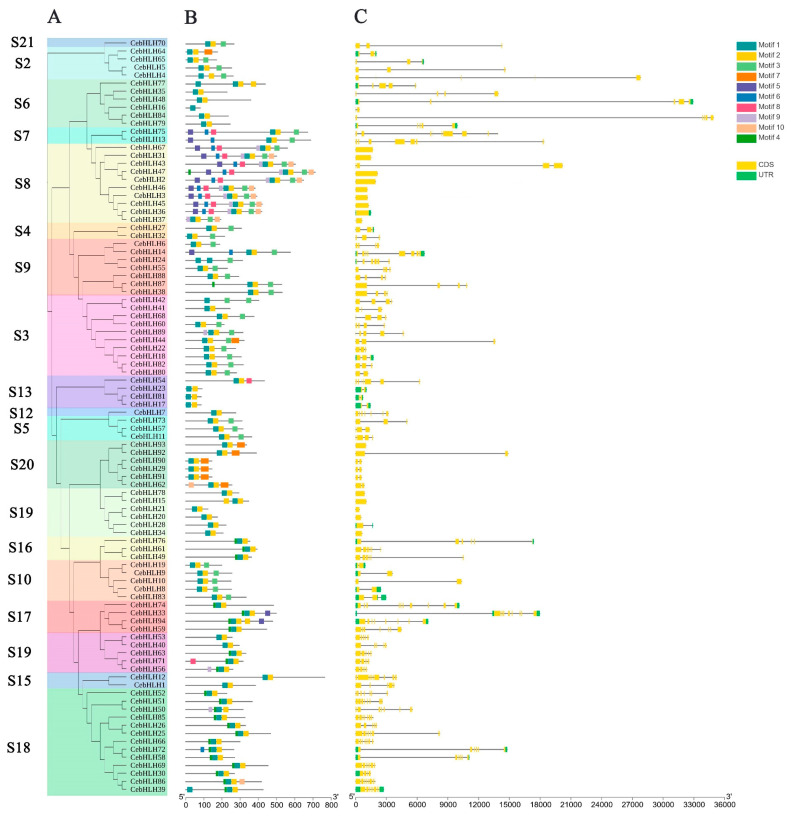
Gene structure and conserved motifs of CebHLHs. (**A**) The phylogenetic tree containing 94 CebHLHs. (**B**) Conserved motifs of CebHLHs represented by squares of different colors. (**C**) Intron–exon structure of *CebHLHs*. The x-axis in B is the number of amino acids (aa), and the x-axis in C is the number of base pairs (bp).

**Figure 3 ijms-24-03825-f003:**
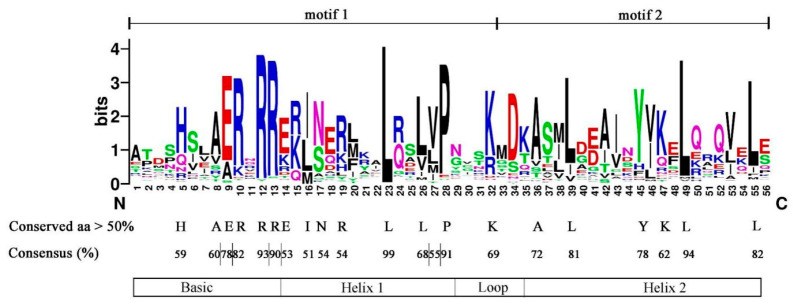
The bHLH domain of CebHLHs, consisting of motif 1 and motif 2.

**Figure 4 ijms-24-03825-f004:**
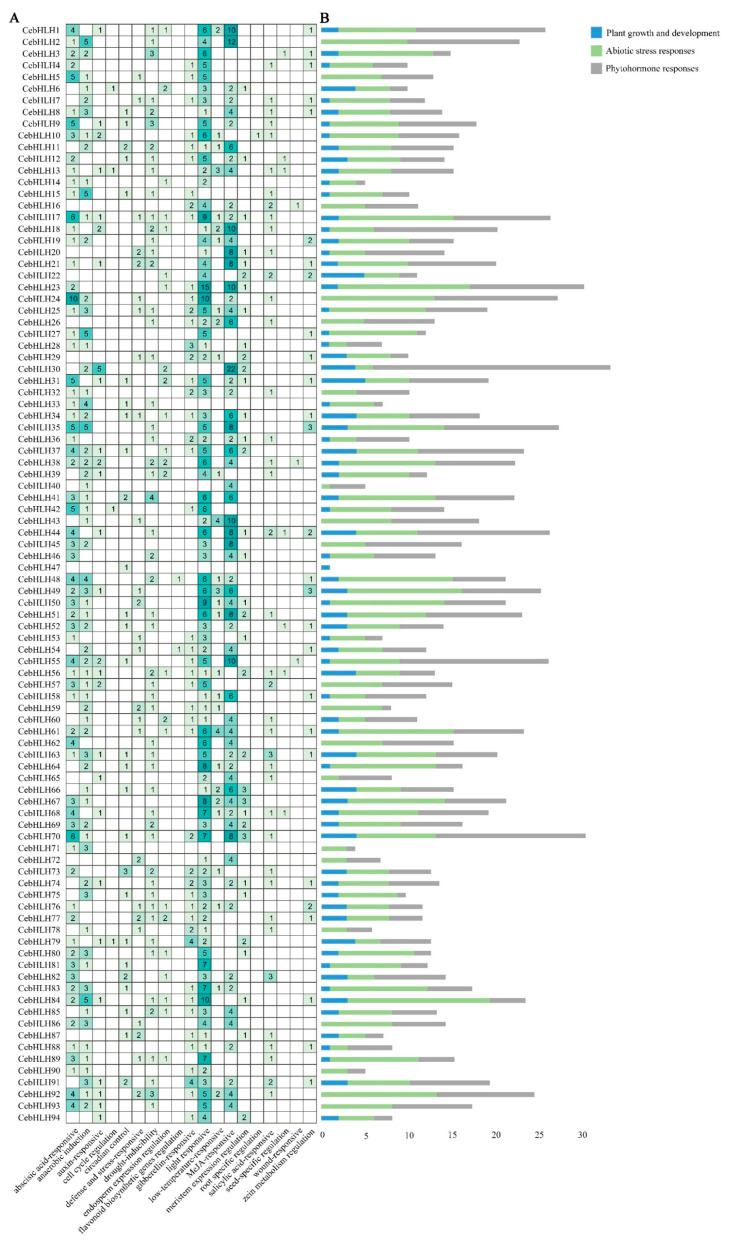
*Cis-acting* elements of *CebHLHs*. (**A**) Number of *cis-acting* elements in *CebHLHs*. (**B**) Blue, green, and gray colors represent the respective three major categories of *cis-acting* elements.

**Figure 5 ijms-24-03825-f005:**
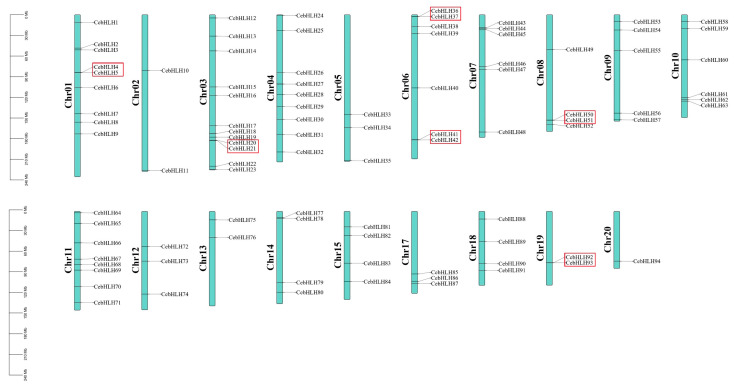
Chromosomal localization and gene duplications of *CebHLHs*. The tandemly duplicated genes are represented by red boxes. The scale bars on the left are the length (Mb) of the chromosomes of *C. ensifolium*.

**Figure 6 ijms-24-03825-f006:**
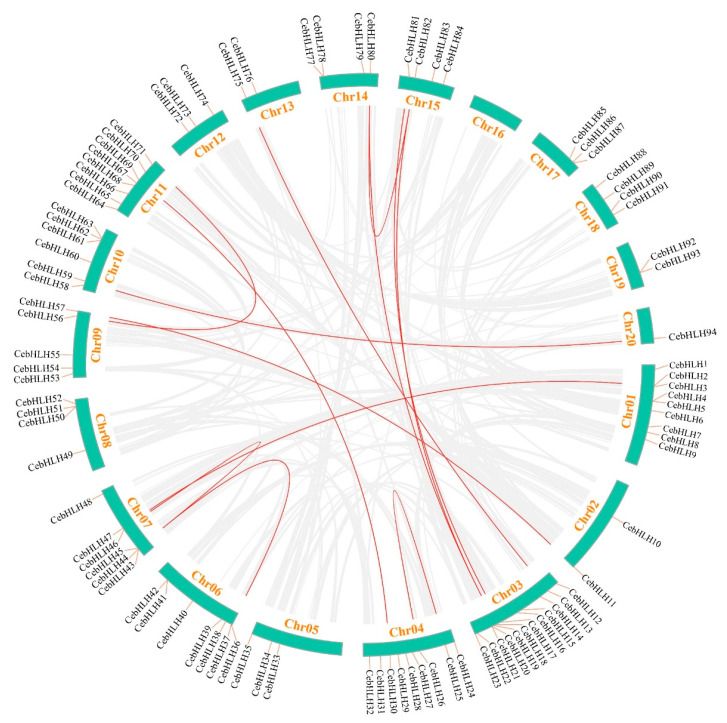
Collinearity analysis of *CebHLHs*. The duplicated gene pairs in the genome were linked by red lines.

**Figure 7 ijms-24-03825-f007:**
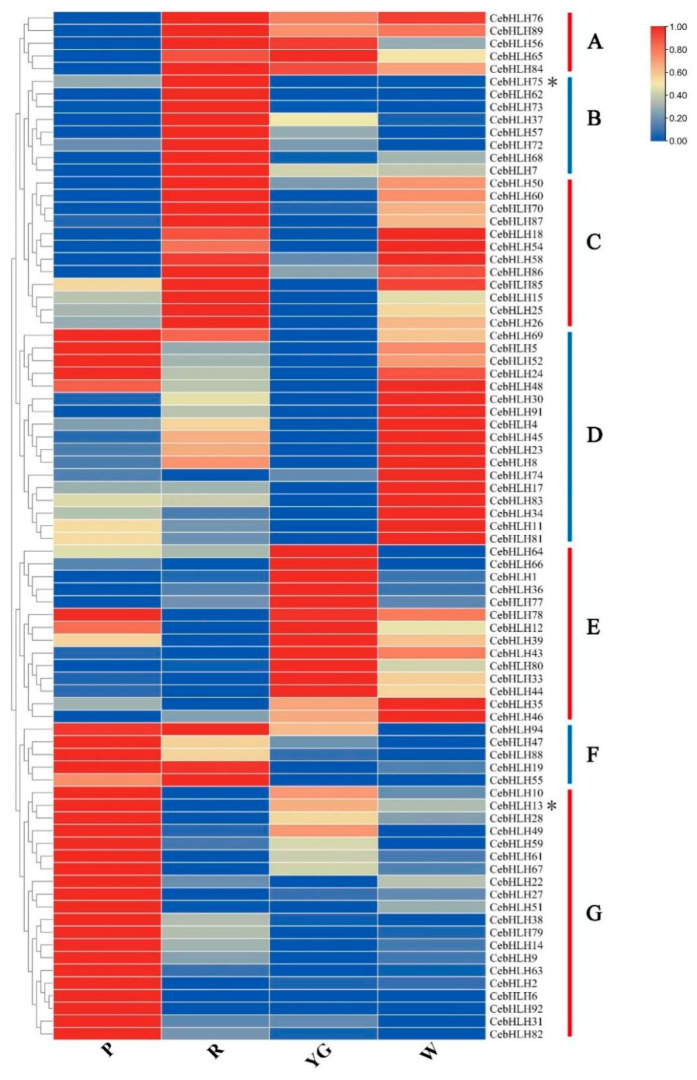
Expression patterns of 82 *CebHLHs* were divided into seven groups (A–G). P stands for purple-red sepals; R stands for red sepals; YG stands for yellow-green sepals; W stands for white sepals. * denotes *CebHLH13* and *CebHLH75* of the S7 subfamily.

**Figure 8 ijms-24-03825-f008:**
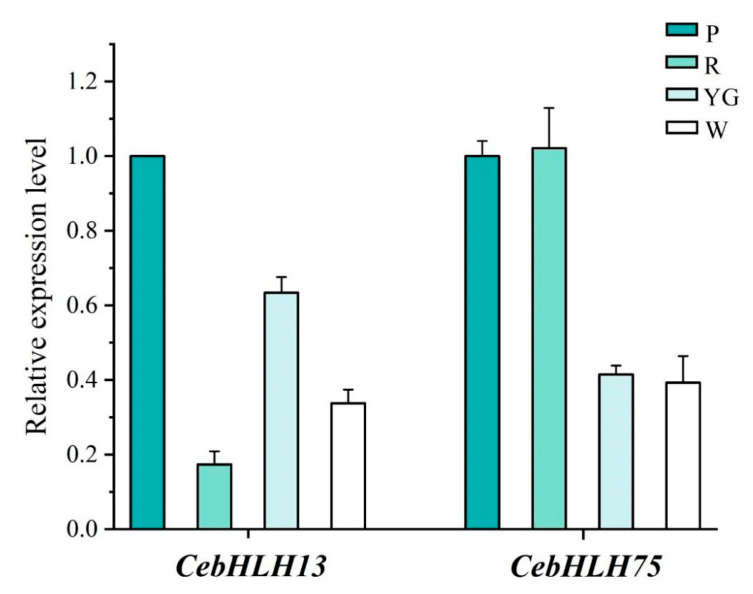
qRT-PCR Analysis of *CebHLH13* and *CebHLH75*. P stands for purple-red sepals; R stands for red sepals; YG stands for yellow-green sepals; W stands for white sepals.

**Figure 9 ijms-24-03825-f009:**
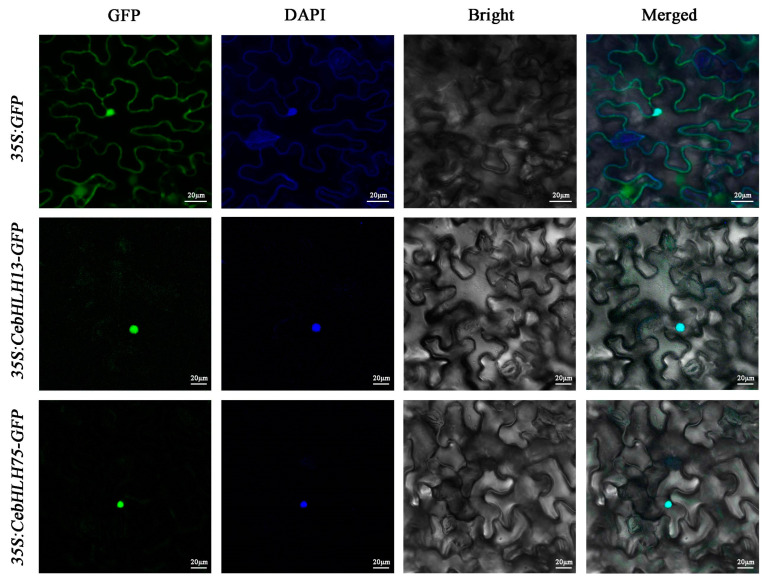
Subcellular localization analysis of CebHLH13 and CebHLH75 proteins.

## Data Availability

All sequences of *C. ensifolium* in this study can be found at the National Genomics Data Center (NGDC). The transcriptomic data are openly available from the National Center for Biotechnology Information under the accession number PRJNA771426. Additional data supporting the article can be found in the Appendix A.

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
