# Peer review of "Genome-Wide Identification and Analysis of bHLH Transcription Factors Related to Anthocyanin Biosynthesis in Cymbidium ensifolium"

_ijms, 2023, doi:10.3390/ijms24043825_

Round 1

Reviewer 1 Report

In this article "Genome-wide Identification and Analysis of bHLH Transcription Factors Related to Anthocyanin Biosynthesis in Cymbidium ensifolium" it is described an extensive search of bHLH factors with the main aim to find some of them, relaterd with flower colour determination.

In my opiion the work is well decribed, the objective are clear and the research design is appropriated.

However, I would suggest to improve a little bit this work by adding some biochemical data to support the conclusions. I'll further detail my few concerns about the work.

1- In figure 2, is the x-axis nt number? Please specify. Is it the study of CDS motifs realized in the CDS?

2- Please explain better the Ka/Ks ratio, the meaning, relevance and consequences of be > o < than 1

3- Localization study in N.  benthamiana leaves need to be confirmed by western blot. In some occasion a nuclear signal could be due to partial degradation of the GFP in the fusion protein. You should check that the size of the fusion `protein after expression in benthamiana is the correct and not partial GFP is present.  Also I observe in the merged field that some cytosolic signal is present but not in the Green - filed. Please check it.

4- Authors speculate that the bHLH factors could homo or hetero- dimerize based on the presence of some conserved amino acids in the bHLH sequence. Because the authors already have a construct with GFP fusion for at least 2 bHLH factors, I would suggest to test in-vivo by "fishing" with the GFP-fusion proteins the total protein extract, and look for some interactors.

At the current state, in my vision the work could be accepted for publication, after revising the concerns exposed.

Reviewer 2 Report

Authors have identified bHLH transcription factors from C. ensifolium and characterized them using different tools. They also studied the role of two bHLH TFs in anthocyanin biosynthesis using qRT-PCR. Authors have written the manuscript well and have provided all the necessary information.

Few minor corrections or suggestions are mentioned below. Authors are advised to check the tense of the manuscript and also do a proper spell-check.

Keyword genome is not appropriate so please remove it. Something like expression analysis can be used instead

Line 43: homologous could be replaced by homodimers

Line 53: is it lips or leaves?

Line 62: ‘also’ is misspelled

Line 93: Please refer Fig. 2a here where 18 subfamilies are mentioned.

Line 95: for S10 subfamily, please mention the number of CebHLH TF number in the parenthesis

In Figure 7 footnotes, please mention what does P, R, YG, W stand for.

Line 215, N. benthamiana needs to be italicized

Line 224, replace ‘genomic’ with ‘genome’

Line 227, Citrus sinensis (56 members) has less number of bHLH than C. ensifolium. So please check the sentence again.

In the discussion section, it would be good to include some information of the A. thaliana bHLH factors from the S7 subfamily because that is the basis for the selection of CebHLH13 and CebHLH75 gene for further characterization.

In the materials and methods section, briefly describe how 94 TFs were identified and what was the source of the reference genome used.

Please provide the deposition ID for the transcriptomic data

Please provide the gene ID for GAPDH. Was the GAPDH gene from C. ensifolium?
